# The Accuracy of Distal Clavicle Fracture Classifications—Do We Need an Amendment to Imaging Modalities or Fracture Typing?

**DOI:** 10.3390/jcm11195638

**Published:** 2022-09-24

**Authors:** Evi Fleischhacker, Georg Siebenbürger, Johannes Gleich, Wolfgang Böcker, Fabian Gilbert, Tobias Helfen

**Affiliations:** Department of Orthopaedics and Trauma Surgery, Musculoskeletal University Center Munich (MUM), University Hospital, Ludwig-Maximilians-University (LMU), 81377 Munich, Germany

**Keywords:** distal clavicle fracture, modified Neer classification, MRI, conservative treatment, operative treatment

## Abstract

Background: Despite its fair-to-moderate reliability, the “modified Neer classification” is widely accepted and used. The purpose of this study was to reevaluate its applicability. Methods: Of *n* = 59 patients with distal clavicle fractures, fractures were classified on standard radiographs. Afterwards, an MRI examination was performed, and fractures reclassified. The primary outcome parameter was quantifying the rate of misclassification. The secondary outcome parameters were the evaluation of the ligamentous injury constellations. Results: In all cases, the fracture course and ligamental integrity could be assigned to the fracture type. Correction of the classification was necessary in *n* = 5 (8.5%) cases. In *n* = 3 (5%) cases, a correction was necessary from Neer I to Craig IIc and thus from conservative to operative treatment. Mean coracoclavicular distance (CCD) in Neer I was 10.2 ± 2.1 mm versus 14.2 ± 3.9 mm in Craig IIc (*p* = 0.02). The mean fracture angle in Neer I was 25.1 ± 3.3° versus 36.8 ± 4.4° in Craig IIc (*p* = 0.02). Conclusion: Cross-sectional imaging resulted in higher precision. Nevertheless, recommendations remain for standard radiographs. The CCD and fracture angle should be considered. An angle of >30° can be assumed as a parameter of instability. A previously undescribed fracture type does not seem to exist. The modified Neer classification is an appropriate and complete fracture classification.

## 1. Introduction

Up to 28% of clavicle fractures involve the distal clavicle [1,2,3]. In 1960, Charles Neer examined the incidence of clavicle nonunion in the context of disruption to the coracoclavicular ligaments [4]. This publication described three fracture types in relation to the lig. conoideum and lig. trapezoideum and is considered the cornerstone of fracture classification in this area. A subsequently published revision of Neer’s classification included avulsion fractures of the periosteal sleeve in pediatric patients and comminuted fractures with an inferior bony fragment attached to the coracoclavicular ligaments [1,5,6]. Edward V. Craig subdivided Neer Type II fractures according to their relation to the lig. conoideum and lig. Trapezoideum [7]. Both adjustments compose the “modified Neer classification,” which is commonly referenced today.

Additional classifications followed over the course of time. For example, Jaeger and Breitner gained great recognition in German-speaking countries in 1984, although they did not introduce a new type of fracture configuration [8]. At the same time as Neer published his classification, Allman classified the entire clavicle, lacking a detailed description of the lateral third [9]. Robinson’s classification from Edinburgh, on the other hand, did not cover coracoclavicular ligament integrity in relation to fracture patterns [3]. The Orthopaedic Trauma Association (OTA) extended its classification system to include the clavicle as bone “15” in the further course [10]. What they all seem to have in common is either a low focus on the distal clavicle and/or poor interobserver reliability [11,12]. Despite the lack of alternatives in principle, it has been the authors’ observation that the modified Neer classification remains the predominantly cited classification system.

The main objective of Neer was to describe the stability of a distal clavicle fracture in relation to the anatomic insertion of the coracoclavicular ligaments. By predicting the stability of the fracture fragments, conservative versus operative treatment can be determined. During the development of the (modified) Neer classification, only X-ray imaging has been available; radiographs represent the gold standard of imaging diagnostics in distal clavicle fractures. The significance of CT and MRI imaging seems to be of little relevance in the diagnosis of lateral clavicle fractures thus far; however, they might provide additional information in the diagnosis of concomitant injuries [13].

Although the modified Neer classification has been used predominantly, focusing on the relationship of the fracture to the coracoclavicular ligaments [14], ligament integrity has never been demonstrated on MRI. Therefore, the purpose of the present study was to reevaluate the modified Neer classification, particularly with regard to the integrity of coracoclavicular ligaments. Where does the inaccuracy of the modified Neer classification come from? Is this due to new fracture types that have not been considered so far? Or is conventional radiographic imaging, always postulated as the gold standard, insufficient and cross-sectional imaging the favorable new diagnostic method?

## 2. Materials and Methods

This prospective study was conducted after the approval of the Ethics Commission of the Ludwig-Maximilians-University (#20-1015). Between February 2020 and April 2022, *n* = 76 consecutive patients with a distal clavicle fracture were screened for eligibility for this trial at our institution (Level-1-Traumacenter). Bony pathologies of the upper extremity, anamnestic prior shoulder surgery (available medical records, scars or third-party medical history) and signs of significant AC-joint arthrosis were considered as exclusion criteria. Also, a ≥72-h delay in MRI examination was considered a delay in treatment and resulted in study exclusion. Fracture displacement was identified on standard radiographs in the anterior-posterior as well as in Rockwood’s view and was classified (Time [A]). Predefined classification was performed according to the modified Neer, Jaeger and Breitner as well as Robinson Classification. Depending on the classification, the fractures were graded into stable (conservative treatment) and unstable (operative treatment). An overview of the classifications used is given in Figure 1. The classification was made by two independent consultants in trauma and orthopedics. Written informed consent was obtained after study inclusion. Subsequently, an MRI examination was performed, and the fractures were reclassified (Time [B]). The primary outcome parameter was quantifying the rate of misclassification between [A] and [B]. The secondary outcome parameter was the assessment of the ligamentous injury constellations in relation to the bony injury based on the MRI images.

Data was enrolled through Microsoft Excel 2010 (Microsoft, Redmond, WA, USA), followed by a statistical analysis using IBM SPSS Statistics, version 25 (SPSS, Chicago, IL, USA) using the Kolmogorov–Smirnov test, Mann–Whitney test and Fisher test. The data were given in terms of the arithmetic mean and standard deviation. Rational data are described by mean and standard deviation. A *p*-value ≤ 0.05 was considered significant for differences in group comparison. A sample size estimation yielded the necessary *n* = 50 patients (margin of error 5%, confidence level 95%) in order to draw conclusions about ligamentous MRI accuracy of 93% with regard to ligamentous injuries of the shoulder [15]. 

## 3. Results

Of *n* = 76 patients presenting with a distal clavicle fracture at our hospital, *n* = 59 patients could be included. The reasons for non-inclusion were a lack of patient consent (*n* = 6, 7.8%) or delayed availability of an MRI examination (*n* = 11, 14.5%). No drop-out occurred after inclusion. The mean age was 38.7 ± 13.2 years, *n* = 12 (20.3%) patients were female. The distribution of fracture classifications at [A] and [B] is presented in Table 1.

In all cases, the constellations of the fracture course and ligamental integrity could be assigned to a fracture type of the modified Neer classification. After reevaluation of the fracture type at time [B], a correction of the classification from time [A] was necessary in *n* = 5 (8.5%) cases. Therefore, in the *n* = 1 (1.7%) case, a correction was necessary within the “unstable” group from Neer IIb to Neer IIa. In *n* = 3 (5%) cases, a correction was necessary from Neer I to Craig IIc and thus from a conservative treatment recommendation to operative treatment. Furthermore, a change occurred in *n* = 1 (1.7%) case of the “stable” group from Neer I to Neer III (see Table 1).

Images A1 and A2 of Figure 2 show exemplary X-rays of the correct fracture configuration of a type I fracture according to the modified Neer classification. Images B1 and B2 show the X-rays of a 28-year-old male. The initial classification was Neer I. An MRI (C1-3) showed a ruptured conoid ligament as well as a ruptured trapezoid ligament. The acromioclavicular capsule and ligament were intact. Thus, by definition, this is an unstable fracture of type Craig IIc.

Images A1 and A3 of Figure 3 show the exemplary X-rays of the correct fracture configuration of a type IIb fracture according to the modified Neer classification. Images B1 and B2 show the case of a 19-year-old female whose initial classification was Neer IIb. An MRI (C1–C3) shows an intact conoid and trapezoid ligament attached to the distal fracture part. The acromioclavicular capsule and ligament are intact. Thus, by definition, the injury type is Neer IIa.

Mean coracoclavicular distance (CCD) in Neer I types was 10.2 ± 2.1 mm versus 14.2 ± 3.9 mm in Craig IIc (*p* = 0.02). The authors came to the recommendation that this value can also be used for differentiation between Neer I fractures and more unstable fracture types, for example, Craig IIc.

Additionally, the authors evaluated the angle of the fracture fragments relative to each other in Neer I and Craig IIc fractures. Figure 4 shows two examples of Neer I versus Craig IIc fracture fragment angles. For evaluation, in the anterior-posterior view, a line was drawn across the proximal portion of the clavicle, from the ligamental insertion area to the fracture. A second line was drawn across the acromion and distal clavicle parts. The mean angle in Neer I fractures was 25.1 ± 3.3° versus 36.8 ± 4.4° in Craig IIc fractures (*p* = 0.02).

Operative treatment strategy of unstable fracture types is shown in Figure 1. The choice of surgical procedure was made according to the current recommendations in the literature. For fracture types located in or distal to the coracoclavicular insertion area (*n* = 20 (50%), Craig IIc and Neer IIb, Jaeger/Breitner IIa), plate osteosynthesis with coraco-clavicular augmentation (Dog Bone^TM^, Arthrex, Naples, FL, USA) was performed in most cases. *n* = 1 (2.5%) case was treated with plate osteosynthesis without coraco-clavicular augmentation, *n* = 1 (2.5%) case with a hook plate. Neer V fractures (*n* = 4, 6.8%) were treated half with plate osteosynthesis with coraco-clavicular augmentation and half with plate osteosynthesis with open fiber cerclage. The majority of Neer IIa fractures (*n* = 10, 71.4%) were treated with plate osteosynthesis. *n* = 3 (21.4%) of these fractures were treated with plate osteosynthesis with coraco-clavicular augmentation and *n* = 1 (7.1%) with a titanium elastic nail (TEN).

## 4. Discussion

Fractures to the distal third of the clavicle represent 10–30% of all clavicle fractures. Neer already recognized a high rate of symptomatic non-union under conservative treatment. For this reason, different fracture classifications have been implemented over the last few decades. The most differentiated and widely used classification up to now is the modified Neer classification, followed by Jaeger and Breitner. Nevertheless, they are under discussion and have limited utility in guiding the treatment of distal clavicle fractures. To date, there have been isolated reports of atypical, not depicted, fracture configurations [16].

The present study attempted to investigate this problem using two separate questions. MRI imaging was the basis of the study, both to find previously undescribed fracture types and to reevaluate the accuracy of the established classification systems.

No previously undescribed fracture constellation was found in this study. Neither in the shape of the fracture nor in the ligamental integrity constellation. This led the authors to conclude that the modified Neer classification is fully appropriate for explaining all distal clavicle fracture types, as well as for determining treatment. However, it cannot be excluded that a new fracture configuration might be found in a larger collective.

Regarding the classification process described in the literature, the authors were able to anticipate a deficiency. Using standard radiographs in anterior-posterior as well as in Rockwood’s view as the gold standard, the diagnostic failure rate for fracture classification was noteworthy (*p* = 0.06). The peculiarity here lies in the accumulation of diagnostic failure in Neer I fractures. The misclassification of this fracture type has clear therapeutic relevance. In most misclassified cases, treatment recommendations changed from conservative to operative treatment due to instability aspects of the involved ligamental injuries. At this point, the question must be asked whether a change in diagnostic standards toward cross-sectional imaging is necessary, which would mean that the modified Neer classification is useful, but the standard diagnostic methods performed are not sufficient. In contrast, there would be the relevant problem of the availability of cross-sectional imaging, as shown in the excluded study collective. Waiting for an MRI can regularly lead to delays, especially in surgical therapy. This problem could perhaps be compensated for by using stress radiographs, although most patients would probably not tolerate this procedure in the acute phase because of pain.

Despite these results, the authors believe that cross-sectional imaging is not necessary as the gold standard. As observed in acromioclavicular-joint injuries, the regular coracoclavicular distance (CCD) is between 11–13 mm and there should be no greater than 5 mm difference between both sides in anterior-posterior view [17,18]. Additionally, the difference in the fracture angles of Neer I and Craig IIc fractures was indeed significant and might be a tool for distinguishing these fracture types. Although the collective was too small to be reliable, larger case series are required to define a cut-off angle. However, at present, an angle of >30° can be assumed approximately as a parameter of instability.

Because of the prospective study approach, there was a disparity in fracture entities. Nevertheless, the prevalence of the individual fracture types reflects the reality of the frequency of fracture pattern [19]. MRI was performed in total with *n* = 10 different MRI machines. The minimum requirement was 1.5 Tesla coils; most of all images were performed with 3 Tesla machine coils. Despite this heterogeneity, the authors recognized no limitations in the assessment and quality of the images. 

## 5. Conclusions

In summary, MRI resulted in a higher precision of fracture classification regarding ligamentous integrity in this study. Nonetheless, the recommendation remains with the standard radiographs in the anterior-posterior as well as in Rockwood’s view. Especially in differentiation of Neer I fractures versus primarily similar appearing, but unstable fracture types, CCD and fracture angle should be evaluated and considered. A previously undescribed fracture type did not exist in this study collective. Applied differentially, the modified Neer classification is an appropriate fracture classification.

## Data Availability

Not applicable.

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
