# Peer review of "The Accuracy of Distal Clavicle Fracture Classifications—Do We Need an Amendment to Imaging Modalities or Fracture Typing?"

_jcm, 2022, doi:10.3390/jcm11195638_

Round 1

Reviewer 1 Report

In general, there are too many typos across the manuscript. I have read this manuscript over and over again, but it's too hard to understand the message that the authors are trying to convey.

Lines 50 : needs references. 

Too many paragraphs in both introduction and discussion sections.

Table 1. Some modified figures describing changes between Time A and B is needed. The graphs 1 and 2 are difficult to easily understand.

Conclusion : Also, too long

Reviewer 2 Report

Thank you for the opportunity to review this article, which focused on the critical aspects of distal clavicle fracture classifications.

The topic is always very relevant. The study is well conducted and the article well written. But some methodological changes are necessary in my opinion:

1) the introduction is a bit too long;

2) an image summarizing the classifications discussed in the article would be useful;

3) the aim of the study should be better explained at the end of the introduction;

4) the methods do not mention the type of statistical analyses performed;

5) results, discussion and images are adequate in my opinion;

6) the conclusions are not an abstract, please do not repeat concepts already treated in the body of the article, just briefly describe the results obtained and summarize the significance of these findings.

Thank you.

Reviewer 3 Report

This is an interesting and well conducted study which is also very well written.

Line 47: Is the low interobserver reliability a claim or ist there literature to back it up

Line 64: The study of Marin Fermin et al. only included studies with arthroscopy to find coocomitant injuries. Because of incdience about 17% with 84% of them needing treatment they recommend MRI. Still its not evaluating the use of a MRI so please rephrase the sentence.

Line 103: Did you search for concomitant injuries?!

Line 131-134: Same information as provided under Fig.1 reference

Line 191-193: Your data of 59 patients might not allow that typ of statement. At least a limitation that an unknown injury might be found with more cases should be made.

205-203: Stress radiographs could determinate the difference between Neer I and Craig IIC and would be cheaper and faster than MRI. Please discuss!

Line 208: If MRI helps to better classify and treat and show concomitant injuries shouldnt it be standard diagnostic, and if yes can you show me one study were delayed surgery of lateral clavicle fractures of about 1-2 weeks changed the outcome?

Line 222: Conclusions is half as long as the discussion and should be shortened.

Round 2

Reviewer 2 Report

The authors have addressed all my concerns. I think the article is worthy of publication in its current form. Thank you.